# Interface Illusions: Uncovering the Rise of Visual Scams in Cryptocurrency Wallets

## ABSTRACT

Cryptocurrencies, while revolutionary, have become a magnet for malicious actors. With numerous reports underscoring cyberattacks and scams in this domain, our paper takes the lead in characterizing visual scams associated with cryptocurrency wallets—a fundamental component of Web3. Specifically, scammers capitalize on the omission of vital wallet interface details, such as token symbols, wallet addresses, and smart contract function names, to mislead users, potentially resulting in unintended financial losses. Analyzing Ethereum blockchain transactions from July 2022 to June 2023, we uncovered a total of 24,901,115 visual scam incidents, which include 3,585,493 counterfeit token attacks, 21,281,749 zero-transfer attacks, and 33,873 function name attacks, orchestrated by 6,768 distinct attackers. Shockingly, over 28,414 victims fell prey to these scams, with losses surpassing 27 million USD. This alarming data underscores the pressing need for robust protective measures. By profiling the typical victims and attackers, we are able to propose mitigation strategies informed by our findings.

## KEYWORDS

cybercrime, scam, cryptocurrency wallet, phishing, visual scam

## 1 INTRODUCTION

The blockchain market is a vast field that encompasses various applications, such as digital currencies, smart contracts, and decentralized applications. According to Fortune business insights [25], the global market size reached $11.14 billion in 2022 with a projected market size of $469.49 billion by 2030. However, as the market expands and becomes more complex, the risks and vulnerabilities of cryptocurrency scams are increasingly evident.

Meanwhile, cryptocurrencies have attracted extensive attention from attackers. CipherTrace reports that total losses exceeded $681 million due to major hacks, thefts, and frauds up to July 2021 [10]. In February 2022, cryptocurrency exchange platform Wormhole lost $320 million after a cyber attack [35]. Recently, emerging scams are also profiting from cryptocurrencies such as the BitConnect Ponzi scheme that resulted in billion-dollar losses [3, 9] and Squid coin scam where fraudsters solicited investments by using the name of "Squid Game", netting $3 million in profits [36].

Traditional cryptocurrency scams have been studied deeply [22, 38, 47, 50], however, a burgeoning category remains underexplored: the visual scams associated with cryptocurrency wallets, a cornerstone of Web3 [15]. This scam exploits the absence of crucial wallet interface details—token symbols, wallet addresses, and smart contract function names—to deceive users, misleading victims into purchasing fake tokens, initiating transactions to attackers, and triggering unintended smart contract function calls (Section 2).

This paper takes the first step to characterize the visual scams of cryptocurrency wallets. We expose three forms of visual scams: Counterfeit Token Scam, Zero-Transfer Scam, and Function Name Scam (Section 3). By analyzing 2,542,283 blocks, encompassing 439,890,433 ERC-20 transactions between July 2022 and June 2023, we identify 24,901,115 visual scam incidents orchestrated by 6,768 distinct attackers. Our analysis identifies that over 28,414 victims were defrauded, resulting in losses exceeding 27 million USD.

To further illuminate the ecosystem of these emerging visual scams in cryptocurrency wallets, we conducted a comprehensive study (Section 4). Our objectives were to identify the cryptocurrencies favored by attackers, determine which user demographics are most susceptible, and understand the strategies employed by scammers.

Our findings indicate that USDT, USDC and ETH are the prime target for scammers, attributed to their high popularity. Notably, users with a shorter registration duration faced the brunt of the attacks, with Counterfeit Token Scam predominantly targeting them. Furthermore, Zero-Transfer Scam resulted in losses that were 60.8% higher than Function Name Scam. Intriguingly, our research unveiled that scammers utilize shared toolkits to launch their attacks. In terms of revenue, we observed a correlation between the frequency of attacks and the revenue: when the profit from attacks is high, attackers flood in; when the operational costs become steep, the attackers tend to sheathe their toolkits.

Based on our quantitative measurement results, we have reported the scammer and toolkit addresses to Etherscan [16] and received their acknowledgment. Besides, we also propose possible mitigation approaches for cryptocurrency wallets, such as educating new-coming wallet users, balancing the security-critical information and UI design, and integrating effective real-time detection methods, to mitigate such scams (Section 5).

**Contributions**. We summarize the contributions as follows:

• We introduce the first longitudinal measurement study for visual scams in cryptocurrency wallets, identifying 24M scam incidents executed by 6,768 attackers.

• We discover the unique ecosystem of visual scams in cryptocurrency wallets, revealing the profile of scam tokens, victims, and scammers and evaluating their profit gains of such scams.

• We propose possible mitigation approaches for cryptocurrency wallets, such as educating new-coming wallet users, balancing the security-critical information and UI design, and integrating effective real-time detection methods, based on our quantitative findings.

## 2 MOTIVATING EXAMPLE

Cryptocurrency wallets like MetaMask [34], Coinbase [11], and Trust Wallet [46] are pivotal interfaces for users to manage their assets. They allow easy transactions such as buying, storing, and transferring. With the rise of mobile computing, wallets have expanded to mobile apps and browser extensions. However, in aiming for user-friendly designs, some wallets may omit "unnecessary" but security-sensitive details, giving scammers a chance to conduct visual scams by exploiting these omissions.

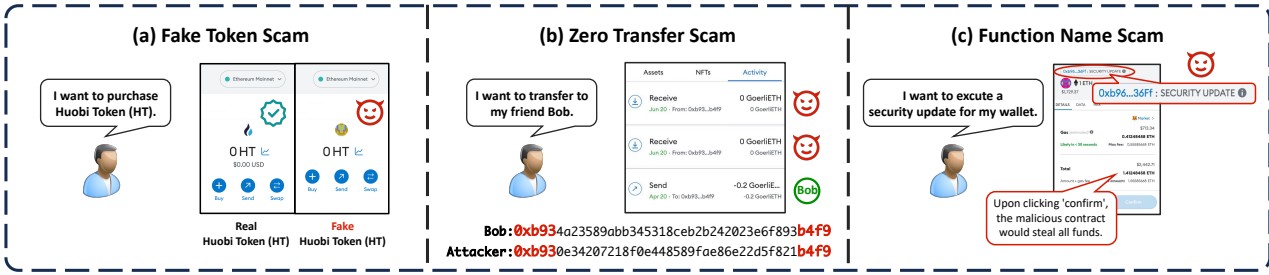

**Figure 1: Motivating Example of Visual Scams**

As shown in Figure 1-a, Alice intends to buy Huobi Token (HT). When presented with a counterfeit token, the wallet still displays the HT symbol. The only discernible difference between the genuine and fake tokens is their logos, which, due to their resemblance, can easily deceive Alice.

In Figure 1-b, when Alice wishes to transfer tokens, the wallet shows a list of recent contact addresses. Alice might choose an address based on matching prefixes and suffixes, assuming it's the right one. However, the wallet's interface omits full address details, making it possible for Alice to inadvertently pick a scammer's address with a similar prefix and suffix, leading to potential losses.

As depicted in Figure 1-c, Alice is misled into upgrading the wallet's security and approves a smart contract labeled `securityUpdate`. The wallet only shows the attacker-configured smart contract function name, concealing that the underlying transaction will transfer Alice's balance to the scammer, thus duping her into the scam.

## 3 DETECTION APPROACH

In this section, we begin by illustrating the mechanisms behind Counterfeit Token Scam, Zero-Transfer Scam, and Function Name scam. Following that, we present the detection methodologies and their corresponding implementations based on their scam logic.

### 3.1 Counterfeit Token Scam

**Scam logic.** In Counterfeit Token Scam, a victim, whom we'll refer to as Alice, intends to purchase or exchange a Huobi Token (denoted as HT). An attacker, however, sends a counterfeit Huobi Token address to her. Despite being counterfeit, the wallet displays the token symbol, HT, which matches the genuine token's symbol, thereby deceiving Alice.

The root cause of these issues lies in the two fundamental attributes specified in the ERC-20 standard: *token name* and *token symbol* [44]. The token symbol usually serves as a shorter version of the name. Despite these attributes act as the primary features for rendering on wallets and identifying cryptocurrency tokens, they can be freely defined by the token's creator during contract deployment. This flexibility allows attackers to deploy counterfeit tokens with attributes identical to genuine ones. Meanwhile, cryptocurrency wallets, for the sake of user-friendliness, display only token attributes like the symbol and name in their user interface, omitting the token address, which differentiates genuine and counterfeit tokens. As a result, the visual similarities between genuine and counterfeit tokens can easily mislead users, leading to significant risks of deception.

**Detection methodology.** To detect counterfeit tokens, we first summarize the most popular forgery methods employed by counterfeit tokens; then we traverse the full ERC-20 token list to discover the counterfeit tokens which satisfy these forgery methods. Specifically, to summarize the forgery patterns, we look at the scam reports [2, 19], and academic researches [22, 29, 39]. Finally, we summarized the following four methods and examples in Table 1.

• **Identical forgery**, where the counterfeit token's name and symbol are identical to the genuine one. For example, a counterfeit token (address `0x6d9952`[1]) has an identical symbol and name to the genuine USDT token (address `0xdAC17F`[2]). The identical forgery has the best phony effects. Due to cryptocurrency wallet and blockchain explorer detecting some malicious identical forgery tokens with warning labels, we witness the attacks also employ the following evasion forgery methods.

• **Cross forgery** involves swapping the name and symbol fields of the genuine token when assigning them to the counterfeit token, *i.e.*, a counterfeit token (address `0x89E894`[3]) has a swap symbol and name to the genuine USDT token. We've noticed a variation of cross forgery where the counterfeit's name and symbol are either both USDT or both Tether USD, like `0x4a401c`[4].

• **Combo forgery** entails adding or removing keywords like "Coin" or "Token" to the fields of the genuine token, ensuring that the semantic meaning of the counterfeit token's attributes remains unchanged, *i.e.*, a counterfeit token (address `0x966657`[5]) has a swap symbol and name to the genuine USDT token.

• **Homograph forgery** involves replacing standard letters with visually similar special characters, *i.e.*, a counterfeit token (address `0xA9ffFc`[6]) using the Ethiopic letter "Ｕ" in place of "U", aiming to make the counterfeit token's attributes visually consistent with the genuine USDT token.

**Implementation.** To begin our comprehensive study, we sourced the ERC-20 token dataset from the Blockchair [5] database, which served as our primary candidate dataset. As of Sept. 26, 2023, we have fetched 799,519 tokens. Drawing inspiration from previous

---

[1] `0x6d995217db76437ea053770dDaB27aA90a298bCa`
[2] `0xdAC17F958D2ee523a2206206994597C13D831ec7`
[3] `0x89E89442Cc2B6e24D43759a7BF5EE1a0029D7BB1`
[4] `0x4a401c912755b2b1e6e486655a74A01c4d455B66`
[5] `0x966657c10A2529Cf7A08B310A13ae0b338B209A6`
[6] `0xA9ffFc9764Ad80362460cb3fb52E53A752053f5d`

**Table 1: Forgery Methods of Counterfeit Tokens**

| Forgery Method | Token Name | Token Symbol | Source |
|---|---|---|---|
| Benign | Tether USD | USDT | |
| Identical | Tether USD | USDT | [19, 22] |
| Cross | USDT | Tether USD | [20, 21] |
| Combo | Tether Token | USDT | [29, 42] |
| Homograph | Tether USD | USDT | [39, 42] |

**Table 2: Account Address Display Patterns of Transfer Records in Mainstream Cryptocurrency Wallets**

| Wallet Client | Display Pattern | Total Users [8] |
|---|---|---|
| MetaMask Extension | 0xaaa...bbbb | 22M |
| MetaMask Mobile | 0xaaaa...bbbb | 10M |
| Coinbase Wallet Mobile | 0xaaaa...bbbb | 10M |
| imToken Mobile | 0xaaaa...bbbb | 1M |
| Trust Wallet Extension | 0xaaa...bbbbb | 1M |
| Trust Wallet Mobile | 0xaaaaaa...bbbbbbbb | 10M |

phishing detection research [33], our study focuses on cryptocurrencies with the highest market capitalization. To this end, we retrieved a list of the top 200 cryptocurrencies from CoinGecko [13]. For each targeted token, we applied the aforementioned forgery methods and excluded genuine tokens verified by CoinGecko and defined by cryptocurrency exchanges [16] to compile the list of potential counterfeits. Subsequently, we examined whether any ERC-20 token matched entries from this list.

Specifically, for identical forgery and cross forgery, we sanitize the name and symbol fields of the tokens, retaining only numbers and letters converted to lowercase for respective matching. For combo forgery, we test all possible scenarios of adding or removing keywords and then proceed with the corresponding match. For homograph forgery, we assess whether the token fields are visually similar to the genuine tokens by substituting with special characters. Given the absence of a complete homograph table in existing research, we constructed one base on Unicode Database [43], and previous related works [41, 51]. The full list of homographs used in our study can be found in Table 5 in Appendix B. Finally, we detect 9,442 counterfeit tokens targeting at the top 200 cryptocurrencies with the highest market capitalization.

## 3.2 Zero-Transfer Scam

**Scam Logic.** We have identified two distinct types of Zero-Transfer Scam, one is crafting victims' recent transaction records by impersonating senders, and another one is crafting victims' recent transaction records by impersonating recipients.

• **Impersonating recipient attack.** A straightforward impersonation this scam exploits the frequent transactions between two parties, dubbed as Alice and Bob, which can be observed in the blockchain. The attacker then initiates the impersonating recipient attack in the following manner: ❶ The attacker generates numerous account addresses and selects an account similar to Bob, denoted as B0b, to act as the impersonating recipient. ❷ The attacker uses the transferFrom function to send a zero-amount from Alice to B0b. Notably, within the ERC-20 specification, if the transfer amount via the transferFrom function is zero, there's no need for authorization from the approve function [44], which allows the attacker to initiate a transfer from Alice without needing Alice's private key. ❸ Above step results in a transfer record from Alice to B0b appearing in Alice's wallet transaction records, which has a similar appearance to the previous genuine records. ❹ Later, when Alice plans another transfer to Bob, she might mistakenly copy B0b's address from her recent transaction records. This error is often due to the truncated display of account addresses on cryptocurrency wallets, where both Bob and B0b appear identically as

"B...b". Consequently, the funds are received by the attacker's account address.

• **Impersonating sender attack.** Another subcategory of Zero-Transfer Scam is the impersonating sender attack. For the context of this attack, let's assume that Alice initially transfers a certain amount of tokens to Bob. The attack unfolds as follows: ❶ The attacker establishes numerous Ethereum address and selects an account that has the same address prefix and suffix to Alice, which we'll denote as Al1ce. ❷ The attacker then uses Al1ce to initiate a zero-amount transfer to Bob. This creates and inserts a transfer record from Al1ce to Bob in Bob's transaction history. ❸ Later, when Bob wishes to transact with Alice, he may mistakenly choose Al1ce as the recipient. This error is facilitated by the truncated/omitted display of account addresses, where both Alice and Al1ce appear indistinguishably as "Al...ce".

**Detection methodology.** The detection methodology for Zero-Transfer Scams can be divided into the following three steps. First, we analyze the display patterns of the popular cryptocurrency wallets and reveal to which degree the impersonating address will have an identical appearance to the genuine one. Then, if Alice received a zero-transaction from B0b, or if Bob received a zero-amount transfer from Al1ce, we take these transfers as zero-transfer attacks. What worse, if Alice makes a non-zero-amount transfer to the impersonating recipient B0b, or if Bob initiates an non-zero-amount transfer back to the impersonating sender Al1ce, we define this transaction as a successful zero-transfer attack.

Specifically, to thoroughly identify impersonating addresses, we examined mainstream wallets spanning both mobile platforms and browser extensions. These include MetaMask [34], Coinbase [11], imToken [23], and Trust Wallet [46]. Subsequently, we initiated an ERC-20 transaction and manually assessed how many bits of the address were obscured by each wallet.

**Implementation.** After examining six mainstream wallets, as shown in Table 2, we find that showing the first three and the last four characters of the account addresses represent the majority of wallets' abbreviation patterns. To locate the impersonating attacks, we devise a detection method. After the timestamp of a normal transfer from Alice to Bob, if Alice make a zero-amount transfer to the impersonating recipient B0b, or if Bob receive a zero-amount transfer from the impersonating sender Al1ce, we define these zero-amount transfers as zero-transfer attacks. Following the attack, if Al1ce or B0b receive transfers from the victim, then the attack is a successful attack.

We first retrieve the list of genuine tokens from CoinGecko [13]. To boost our traversal performance, we deployed an Ethereum full node with Erigon [30]. We traverse through all ERC-20 transfers to obtain all genuine token transfers with a zero-amount, while building hash tables for easy retrieval. For every zero-amount transfer, we determine the presence of zero-transfer attacks and successful attacks based on the method defined above.

## 3.3 Function Name Scam

**Scam logic.** Smart contracts are increasingly utilized in various sectors, including finance, gaming, and other legal industries, to conduct business autonomously without human intervention. Like traditional computer programs, each function in a contract is defined by its name and body. When users engage with these contracts, most cryptocurrency wallets only present the function's human-readable name, instead of the function code.

However, a concerning trend has emerged where attackers assign deceptive function names, such as `securityUpdate` or `claimRewards`, that don't align with the function's code behavior. Instead of executing the expected actions, these malicious functions merely seek the user's authorization to transfer cryptocurrency. By pairing these misleading function names with phishing webpages or messages, attackers can trick users into signing transactions or granting permissions that, concealed from them, drain their funds.

**Detection methodology.** The key to detecting function name scams lies in determining whether the direction of funds flow aligns with the function name semantics. For instance, `securityUpdate` implies funds hold, `claimRewards` indicates funds inflow, while `transfer` suggests funds outflow. Specifically, we focus on transactions where funds are outgoing, even though the function names do not inherently suggest such outgoing transaction semantics.

To retrieve function names with elusive semantics, we employ the snowball algorithm. Initially, we gathered deceptive function names from anecdotal reports. We then iteratively assess function names based on the nearest semantic embedding distance and manually label those with elusive semantics. In the final step, we investigate whether such functions indicate outgoing transactions.

**Implementation.** By examining all Ethereum transactions over a year, we pinpointed 25,982 function names that were actually used. Our initial dive into scam reports [17, 18] highlighted two potentially deceptive function names: `securityUpdate` and `claimRewards`. To expand seed set, we randomly selected 1,000 function names from the above list for manual inspection, which revealed 10 more misleading function names. With these 12 names as seeds, we employed a method inspired by feature propagation [26] to sift through these function names.

Utilizing Google's pre-trained Bert model [14], we extracted feature vectors from function names. For each seed, we identified the three closest function names by Euclidean distance, incorporating them for the next iteration. This cycle repeated two times, culminating in the identification of 156 function names. We manually reviewed these function names and excluded 16 whose semantics suggested fund outflows, leaving 140 potentially misleading function names. During the traversal process of Ethereum transactions, if we identified a transaction that results in funds outflow and its input data aligned with any of the misleading function names, we

determined that this transaction is an instance of Function Name Scam. Finally, we identified 17 distinct scam functions originating from eight deceptive function names, which are fully presented in Table 4 in Appendix A.

## 3.4 Evaluation

In this subsection, we rigorously evaluate the detection methodologies for visual scams. To evaluate the recall of our approach for Counterfeit Token and Zero-Transfer Scams, we gathered scam reports from Twitter, given its prominence as the platform for blockchain-related news. Our method successfully detected 16 of 18 counterfeit tokens, reported in the flowing posts [2, 27, 45]. One missing case (0x75409A[7]) pertained to the token, ETHM, which falls outside the top 200 in market value. Another undetected token (0x5F799A[8]) was designed to forge USDT, with the name "USDT ERC-20" and the symbol "USDT", utilizing a combination of cross and combo forgery methods. We did not consider the combination of various forgery methods, as it could lead to a significant increase in false positives. As for Zero-Transfer attacks, our results successfully captured all 550 attacks, reported by [1, 4, 32].

We also assessed our tool's recall by comparing it to Scam Sniffer, a reputable anti-scam platform [40]. Scam Sniffer has documented instances of Function Name Scam using a blocklist of malicious contracts. Their records indicate 2,513 successful attacks with a combined loss of 1,833 ETH. When evaluating the recall for this scam, we found that our results entirely cover the data from Scam Sniffer. Impressively, our detection highlighted a total of 33,873 successful attacks, 13 times their count.

As for precision, due to the lack of an off-the-shelf ground truth dataset, we randomly selected 100 scams from each scam category and undertook a manual validation. The results show our method can precisely detect Zero-Transfer and Function Name Scams. Only one false positive was spotted in the Counterfeit Token Scam, which address had received "counterfeit GALA" (0x15D4c0[9]). A deeper dive revealed that owing to a contract upgrade, the token had been migrated, signifying that the token in question was the old GALA [12], not a counterfeit token.

## 4 MEASUREMENT

In this section, we conduct a comprehensive study to determine: when scams occur, and which tokens are targeted by these scammers. Interestingly, we also examine the types of victims they primarily target and the toolkits employed by the scammers and estimate the revenue attackers garner from such scams.

### 4.1 Landscape

**Scale.** Our study was conducted on the Ethereum blockchain from July 1, 2022, to June 30, 2023. Within this period, we examined 2,542,283 blocks, encompassing 439,890,433 ERC-20 transactions. Leveraging the detection approach detailed in Section 3, we successfully identified a total of 24,901,115 visual scam attacks, comprised of 3,585,493 counterfeit token attacks, 21,281,749 zero-transfer attacks, and 33,873 function name attacks. According to the results, we

---

[7] 0x75409AC44f95Ce4106336716E47C03dc817cB56a
[8] 0x5F799AD15d02B2668d37575B2fB6eBaeee368A05
[9] 0x15D4c048F83bd7e37d49eA4C83a07267Ec4203dA

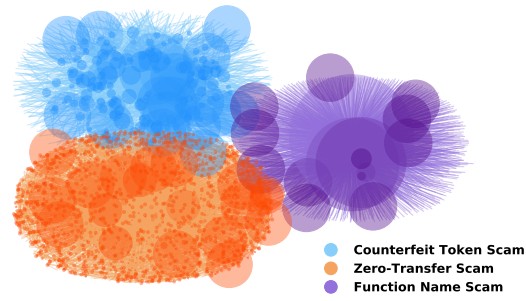

**Figure 2: Real-World Campaign of Visual Scams**

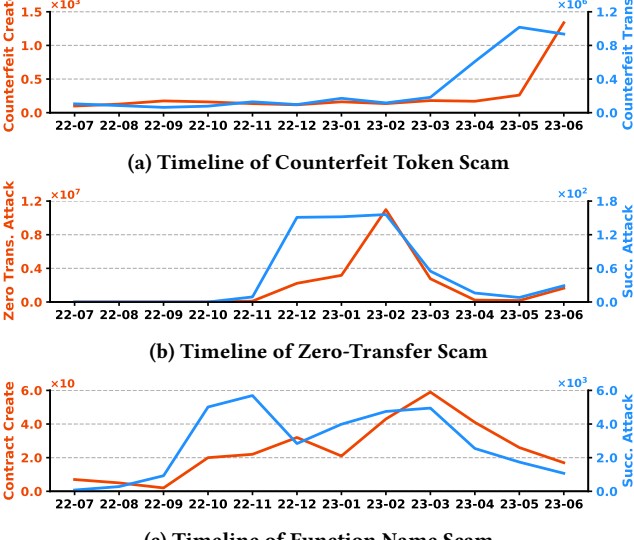

(a) Timeline of Counterfeit Token Scam

(b) Timeline of Zero-Transfer Scam

(c) Timeline of Function Name Scam

**Figure 3: Timeline of Visual Scams**

**Table 3: Distribution of Visual Scams**

| Scam Type | Analysis Dimension | Category | # (%) |
|---|---|---|---|
| Counterfeit Token | Token Types (# of Tokens) | USDT | 2,115 (22.4%) |
| | | ETH | 1,449 (15.3%) |
| | | USDC | 984 (10.4%) |
| | | HT | 653 (6.9%) |
| | | BTC | 401 (4.2%) |
| | Forgery Methods (# of Tokens) | Identical | 3681 (39.0%) |
| | | Cross | 3309 (35.0%) |
| | | Combo | 1713 (18.1%) |
| | | Homograph | 739 (7.8%) |
| Zero-Transfer | Token Types (# of Succ. Attacks) | USDT | 307 (53.3%) |
| | | USDC | 194 (33.7%) |
| | | BUSD | 18 (3.1%) |
| | | DAI | 13 (2.3%) |
| | | QNT | 12 (2.1%) |
| Function Name | Function Names (# of Succ. Attacks) | securityUpdate | 25,442 (75.1%) |
| | | claimRewards | 4,979 (14.7%) |
| | | claimReward | 2,911 (8.6%) |
| | | claimQuestRewards | 528 (1.6%) |
| | | upgradeStrength | 11 (0.03%) |
| | | getBonus | 1 (0.003%) |
| | | upgradeReward | 1 (0.003%) |

determined 5,307 fake token scammers who issued 9,442 counterfeit tokens, 1,193 zero-transfer scammers who utilized 1,057 malicious contracts, and 268 function name scammers who deployed 309 malicious contracts based on 17 illusive functions. In total, these scams affected 2,196,303 Ethereum addresses, resulting in financial losses of 27,359,760 USD for 28,414 victims (See Section 4.4).

**Campaign analysis.** We illustrate the real-world campaign of visual scams in Figure 2. The small dots represent the affected addresses, the lines indicate the occurrence of attacks, and the larger circles denote the scam toolkits initiating the attacks. For Counterfeit Token Scam, several prevalent cryptocurrencies, such as USDT and ETH, are the most frequently impersonated, occupying the larger circles. For Zero Transfer Scam, attackers execute transfers of zero amount to a massive number of wallet addresses. Some wallet addresses, due to their large balances or recent history of high-value transactions, become the attractive target of multiple attackers, presenting as larger circles on the figure. For Function Name Scam, we discovered that the functions with the names of `Sec urityUpdate` and `claimRewards` constitute the largest proportions. Besides, there are several other ambiguous functions contributing to different degrees within this scamming strategy.

**Timeline.** The digital realm has witnessed dynamic shifts in terms of security threats and vulnerabilities over recent years. However, the timeline pattern of the three visual scams varies a lot.

Historically, the creation and transfer volumes of counterfeit tokens have remained at a relatively low level. However, there was a sudden surge in 2023 Q2. In June 2023, counterfeit creations and corresponding transfers were 7.43 (1,337 vs 180) times and 5.12 (935,086 vs 182,668) times, respectively, more than in March 2023.

Zero-Transfer Scam was first identified on TRON network [24]. It didn't take long before Ethereum became a major target, with attacks amplifying in scale from December 2022. Over the next two months, the scale of zero-transfer attacks expanded dramatically, peaking at 5,254,205 attacks in February 2023, with the number of successful attacks reaching its peak of 153 monthly cases. Subsequently, the trend started to decline, but there was a sign of a potential rebound in June 2023.

For Function Name Scam, there were only 75 successful function name attacks in July 2022, but this number soared to 5,693 in November. Throughout the year, two peak periods of Function Name

Scam emerged: the first around November of 2022 and the second around March of 2023, which was larger in scale and duration.

**Distribution.** Table 3 depicts the distribution of the three scams according to their token types, forge methods, and function names. Among the top 200 cryptocurrencies, we witness that 181 have been subject to counterfeiting. Out of the 9,442 counterfeit ERC-20 tokens we detected, 6,353 are designed to imitate legitimate ERC-20 tokens (*e.g.*, USDT and USDC), while 3,089 are modeled after non-ERC-20 tokens (*e.g.*, Bitcoin and Ethereum). USDT tops the list as the most frequently counterfeited token, with a total of 2,115 counterfeits. It's followed by ETH and USDC, which have 1,449

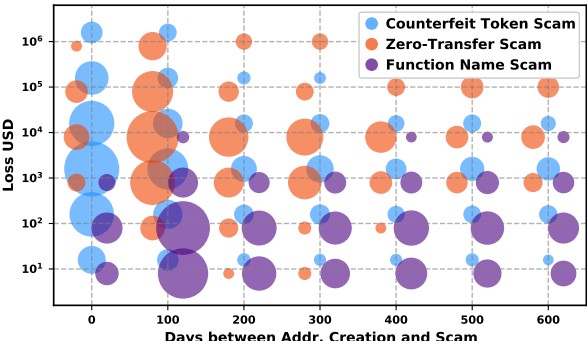

**Figure 4: Temporal and Monetary Analysis of Victim Profile**

and 984 counterfeits respectively. This highlights a notable issue: many victims seem to lack a basic understanding of blockchain, given that ETH and Bitcoin are not ERC-20 tokens, which can not be bought through the Ethereum token transaction.

> **Finding I**: *Many victims of visual scams have a limited understanding of cryptocurrency, as evidenced by their attempts to purchase Bitcoin on the Ethereum blockchain.*

Regarding forgery methods, only 39.0% of counterfeit tokens precisely replicate genuine tokens. This observation underscores a key difference between counterfeit cryptocurrencies and traditional counterfeit currency in the real world, which often aims for a precise imitation. However, 61.0% of ERC-20 counterfeit tokens deliberately diverge from their genuine versions, which indicate that they tend to utilize such difference to bypass possible mitigation applied by cryptocurrency wallets.

Zero-Transfer Scam primarily focuses on high-circulation ERC-20 tokens. Among the most affected are Ethereum's popular stablecoins, with USDT and USDC accounting for 87.0% of the zero-transfer scams. Stablecoins, due to their 1:1 peg with fiat currencies, possess substantial liquidity and volume in the crypto market, often serving as trading intermediaries. Their widespread use positions them as prime attractions for attackers.

We observed that the toolkit of Function Name Scam shows a long-tail distribution. 75.1% of successful attacks were attributed to the function name `securityUpdate`. Following this, three function names related to "reward"—`claimRewards`, `claimReward`, and `claimQuestReward`—together accounted for 24.9% of successful attacks. The remaining function names represented a minor fraction with only 13 successful attacks.

Interestingly, we found that 6.6% of the counterfeit tokens were established before their genuine counterparts, with an additional 0.4% being created on the exact same day as the genuine ones. This indicates that some attackers might be preemptively targeting newly introduced cryptocurrencies based on pre-launch news.

> **Finding II**: *Counterfeit tokens launched by proactive attackers even appeared before the genuine token ICO.*

## 4.2 Victim Profile

Figure 4 provides an insightful victim profile of visual scams based on two metrics: the horizontal axis representing the time elapsed

between the account address creation and the scam event, and the vertical axis indicating the victim's cryptocurrency losses (converted to USD). The size of each circle denotes the number of victims. Notably, for Counterfeit Token Scam, we can not directly understand how much financial losses have been made during the victims receiving such counterfeit tokens. To conduct a conservative estimation, we refer to the method used in previous work—using the value of the corresponding genuine tokens instead [22].

**Temporal analysis.** As depicted in Figure 4, different types of scams target victims based on their address creation time. Counterfeit Token Scams (blue circle) are situated closest to the left of the graph. Remarkably, 39.1% of account addresses received counterfeit token transfers on the first day of their creation. This suggests a significant portion of the victims had no prior experience with blockchain and Ethereum before falling prey to this scam. Furthermore, 59,1% of the addresses that received counterfeit token transfers were newly created wallets within four months, indicating that attackers predominantly target newcomers.

> **Finding III**: *Newly created accounts, particularly those within the first four months, are favored targets for attackers, enduring approximately 60 percent of the scams.*

Zero-Transfer Scam (orange circle) predominantly targets addresses created within roughly 200 days of their inception. Notably, 37.5% of its victims are deceived within their first four months, suggesting that newer cryptocurrency wallet users are more vulnerable to this visual scam compared to seasoned counterparts.

Function Name Scam (purple circle) shows a broader spread across the timeline, suggesting that more "experienced" users might fall victim to this scam. The victim with the highest loss, `0x0E7A6b`[10], an Ethereum user for six years, was scammed by the deceptive function name `claimRewards`, losing the entire balance of 495 ETH, equals to 956,510 USD.

We uncovered a total of 33,873 successful function name attacks, affecting 27,854 victims. Alarmingly, 10.1% of victims fell for function name scams more than once. The account address `0xD256A2`[11] approved three malicious contracts consecutively 21 times within 98 days, which included calling one `claimRewards` function 18 times, one `securityUpdate` function twice, and another `securityUpdate` function once. This behavior indicates that even after multiple deceptions, the user of this address remained oblivious to the significant risk these contracts posed to their assets.

> **Finding IV**: *10.1% of victims were repeatedly scammed by malicious contracts, with someone falling 21 times serially.*

**Monetary impact.** As shown in Figure 4, the loss of scams can be categorized into three folds. Zero-Transfer Scam is the most skewed towards the top of the graph, with an average loss of $30,126 per victim. The victims of Zero-Transfer Scam suffer the highest loss among the three scams. Specifically, among the 560 victims of this scam, 42 victims suffered losses greater than 100,000 USD. We discovered that these victims share common characteristics: they had recently made large transfers, or they had significant balances in their wallet addresses. Given the transparency of blockchain's distributed ledger, attackers selectively target valuable addresses

---

[10]`0x0E7A6b3b5EE4A1228A0334FA8170347A31538c49`
[11]`0xD256A23425B770baB6AF00123a16e387D81D5C00`

by examining on-chain data like transaction histories and account balances. `0x081714D`[12] suffered the most significant loss in this scam, losing 2,030,000 USDC to an impersonating recipient. The victim was deceived only 42 days after the creation of this address and was targeted by 15 zero-transfer attackers simultaneously.

Conversely, Function Name Scam results in the smallest average loss per victim among the three scams, with each victim losing just 377 USD, merely 1% of Zero-Transfer Scam loss. Additionally, 95.9% of victims in Function Name Scam suffered losses below 1,000 USD. Meanwhile, Function Name Scam swindled 27,854 individuals, compared to the 560 victims by Zero-Transfer Scam. This accumulation of small losses sums up, making the total monetary impact of them roughly comparable.

> **Finding V**: *The profit models of scams vary; some accumulate small amounts, while others rely on a few substantial gains.*

## 4.3 Attacker Profile

**Scamming Toolkits.** To understand how attackers orchestrate visual scams, we examined the toolkits used in these schemes. In Zero-Transfer Scam, attackers initiated the illusive transaction by invoking `transferFrom` functions from smart contracts. Since this feature allows multiple transfers simultaneously [6], it makes the scam more efficient and cost-effective. 1,193 attackers carried out these 21,281,749 assaults using 1,057 distinct contracts. An alarming observation was that certain smart contracts, serving as reusable attack toolkits, have been employed by different attackers. For instance, the contract at `0xc46cd1`[13] was harnessed by four different attackers, culminating in 104,137 zero-transfer attacks.

In Function Name Scam, only seven distinct function names are associated with 268 attackers. They targeted 17 deceptive functions related to and deployed 309 unique malicious contracts. A minority of malicious contracts dominate the scale of this scam, with only 9.1% of contracts having more than 100 successful attacks. Among them, five malicious contracts launched over 1,000 successful attacks, collectively accounting for 72.9% of all successful attacks, indicating a significant monopolization in this scam.

**Aggressive perpetrators.** In visual scams, some aggressive attackers launch numerous scams to deceive victims. In Counterfeit Token Scam, attacker `0xEAB4Fb`[14] stands prominent, having crafted 233 distinct counterfeit tokens, impersonating ETH, DAI, BUSD, USDT, and USDC. The `0x4f0622`[15] stands out as the most active perpetrator among counterfeit tokens, whose name and symbol are identical to USDT, which launched an astonishing 580,527 transactions in a mere two days. Within Zero-Transfer Scam, the attacker `0xFfFfe7`[16] emerges as the most active participant, mounting a staggering 384,514 attacks through 15,017 transactions.

Regarding Function Name Scam, the `0xD361e2`[17] is notably the most active attacker, who used the deceptive function name `securityUpdate` to execute 15,683 successful attacks, yielding a profit of $1,888,557. On the other hand, the `0x000000`[18] emerged as the

---

[12] `0x081714D70d61d80b078eF0dC88022E08dD53236E`
[13] `0xc46cd1a4b3D14451F76fda8C33374f8AF749F907`
[14] `0xEAB4Fb43bB45b917bA1Ce0Cb28bdEdC9a4b7d081`
[15] `0x4f06229a42e344b361D8dc9cA58D73e2597a9f1F`
[16] `0xFfFfe71e7e6Bc965712c91b693a75d2bf717FFF0`
[17] `0xD361e29C48841C40506FC6E6211f68a203Ec1Ef1`
[18] `0x00000000001AdC2c0b202D0f72AD9d50F0675296`

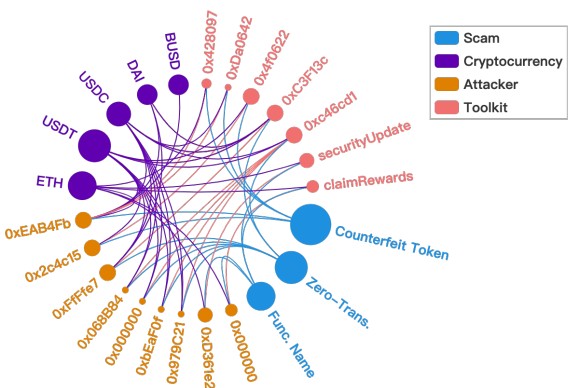

**Figure 5: Interconnections within Attacker Profile**

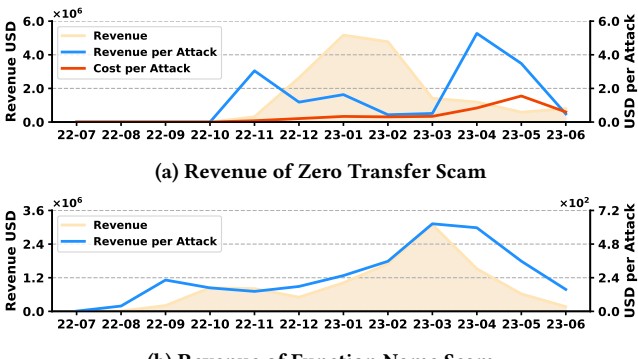

**(a) Revenue of Zero Transfer Scam**

**(b) Revenue of Function Name Scam**

**Figure 6: Revenue of Visual Scams**

highest profitable attacker. By employing the `claimRewards` function name, they conducted 1,744 successful attacks, accumulating a remarkable $2,010,241 in profits.

## 4.4 Revenue Estimation

In this section, we estimate the scam revenue. Even though is almost an impossible mission to cover every real-world scam, we still argue that our method can shed light on the lower bound of how much the attacker can gain from the visual scams.

**Zero-Transfer Scam.** As shown in Figure 6-a, in November 2022, when the zero-transfer attacks first appeared, their novelty caught many off guard, allowing attackers to reap revenues as high as 38.45 times their attack costs. However, as details of this attack were revealed on forums and scam reports as follows [1, 4, 48], the community became more alarmed about such scams. By February 2023, the profitability of these scams fell to just 41.35%. The primary expense for zero-transfer attacks is the gas fee associated with transactions. Ethereum's gas fees reached year-long peaks in April and May 2023 [37], which further deterred scammers.

Notably, during April and May 2023, the profits from individual attacks saw unexpected spikes. This was attributed to two significant losses: address `0x44B6A3`[19] lost a whopping 849,995.5 USDT

---

[19] `0x44B6A393560f9146E7556F0894b4Ce76875B92f4`

in April, while `0x3804d7`[20] mistakenly sent 200,000 USDC to a deceptive address in May.

**Function Name Scam.** The only cost for attackers in Function Name Scam is the gas fee to deploy the malicious contracts onto Ethereum, which is negligible compared to the revenues, especially since the gas fee incurred when a user interacts with the malicious contract is paid by the user themselves [6]. `securityUpdate` stands out as the most lucrative deceptive function name, amassing $5,279,297 from 25,442 successful attacks. Close behind are the similarly titled `claimRewards` and `claimReward`, respectively raking in $4,165,771 and $1,030,711 from 4,979 and 2,911 breaches.

Figure 6-b illustrates the total revenue and revenue per attack. Similar with Figure 3, there are two spikes in the revenue of this scam, with the latter spike significantly surpassing the former. March 2023 marks the top of revenue, during which attackers collected as much as $3,091,997 in total and $625 per attack.

## 5 MITIGATION

In this section, based on our invaluable insights, we propose practical and effective mitigation strategies for cryptocurrency wallets.

**Educating new-coming wallet users.** The naive newcomers to the blockchain are especially susceptible due to visual misdirection from wallets. In Counterfeit Token Scam, 59.1% of victims are newly created within four months. Many victims lack a basic understanding of cryptocurrency, like attempting to buy Bitcoin on Ethereum ERC-20 token. Moreover, 10.1% of victims remained oblivious even after falling prey to Function Name Scam. As a countermeasure, cryptocurrency wallets should offer comprehensive guidance to newcomers, *i.e.*, educating users about prevalent scam tactics, to avoid potential financial losses.

**Balancing the security-sensitive information and UI design.** Nowadays, cryptocurrency wallets tend to omit some less "crucial" information on the UI for user-friendliness. In Section 2, we demonstrate the root cause of visual scams in cryptocurrency wallets stems from the absence of crucial details, *e.g.*, the address, which scammers exploit to swindle victims. We suggest the wallet developers should balance the design and provide detailed information. Specifically, to prevent zero transfer attacks, we recommend displaying at least 10 hex characters of the transaction address.

**Integrating effective real-time detection methods.** One effective countermeasure is to notify users when they're about to engage with a scam contract. However, the leading online Ethereum explorer predominantly depends on user or corporate organization labeling [7]. As highlighted in Section 3.4, many blockchain anti-scam platforms primarily utilize blocklists. Our findings in Section 4.1 revealed that 7.0% of counterfeit tokens emerged either before or concurrently with their genuine counterparts. Given the rise of such proactive attackers, it's imperative for cryptocurrency wallets to adopt real-time strategies that can accurately identify scam transactions and tokens.

## 6 RELATED WORK

**Blockchain scam.** As the ecosystem of cryptocurrencies has expanded, scams aiming to pilfer digital assets have concurrently

risen. Phillips and Wilder [38] highlighted the proliferation of cryptocurrency scams on visually similar websites, which deceive users by leveraging the blockchain's transparency. Cryptocurrency exchange scams have been spotlighted by Xia *et al.* [50]. Their efforts in gathering scam domains and fake apps revealed a financial loss of over 520k US dollars. Li *et al.* [31] identified over 10,000 giveaway scam websites targeting users of popular cryptocurrencies. In the realm of Ethereum smart contracts, Ji *et al.* [28] uncovered the "fake deposit" vulnerability and pointed towards susceptible contracts. Wang *et al.* [47] shifted the focus to malicious browser extensions themed around cryptocurrency, underscoring their evasion of detection. Tapping into the global sentiment, Xia *et al.* [49] characterized cryptocurrency scams that capitalize on the COVID-19 pandemic.

Notably, the existing literature seems sparse on scams specifically targeting cryptocurrency wallets, indicating a potential avenue for further exploration. The work most similar to ours is by Gao *et al.* [22], who identified 2,117 counterfeit tokens targeting the top 100 tokens on Ethereum. In contrast, our research reveals that 32.7% of ERC-20 counterfeit tokens target cryptocurrencies outside of Ethereum, a detail overlooked in their study. Notably, our methods identified 4.5 times more counterfeit tokens (9,442 vs. 2,117). Our analysis of both victim and attacker profiles offers a more comprehensive understanding and mitigation approaches to such scams.

**Web visual phishing.** Visual similarity has recently garnered attention due to its increasing prevalence and advanced deceptive techniques. Attackers exploit various feature dimensions, specifically domain names and webpage appearances, which users often focus on, to execute fraud. To construct fake domains, attackers usually leverage the homograph and combosquatting technique. Homograph domains, which exploit the visual similarities of characters, have been recognized as a phishing tool. Quinkert *et al.* [39] measured attacks of homograph domains. The concept of "combosquatting" was explored in-depth by Kintis *et al.* [29]. Their analysis of over 468 billion DNS records reveals a wide spectrum of malicious activities. Moreover, Tian *et al.* [42] spotlighted squatting phishing domains by visual analysis, revealing over 90% of detected phishing pages bypassed popular blacklists. In the other way, the attacker constructs a webpage with the same appearance as the genuine to attack. Lin *et al.* [33] focused on the visual identification of phishing webpages. To the best of our knowledge, none of the mentioned studies delves into the specific threats of cryptocurrency wallet-based visual phishing.

## 7 CONCLUSION

This paper presents the first measurement study on the visual scams of cryptocurrency wallets, providing a novel and comprehensive lens on this emerging cybercrime. Over a span of one year, we identified an alarming 24.9 million incidents, shedding light on the strategies and toolkits of 6,768 unique attackers. Our analysis reveals that over 28,414 victims were defrauded, resulting in losses exceeding 27 million USD. By discovering the unique ecosystem of visual scams in cryptocurrency wallets, we reveal the flavor profile of victims, and scammers and evaluate their profit gains from such scams. Our research also provides the recommendation of mitigation strategies informed by our findings.

---

[20]`0x3804d78b3966fC47d77D41AE8ee190A2d90f5da7`

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

# A DETAILS OF FUNCTION NAME SCAM

**Table 4: Misleading Function Names in Real-World Scams**

| Function Name | Function | Func. Selector | Succ. Attack |
|---|---|---|---|
| securityUpdate | SecurityUpdate() | 0x5fba79f5 | 25300 |
| | SecurityUpdate(address) | 0x593dae5b | 142 |
| claimRewards | claimRewards(address) | 0xef5cfb8c | 2814 |
| | claimRewards(uint256[]) | 0x5eac6239 | 1412 |
| | ClaimRewards() | 0x12798972 | 486 |
| | claimRewards() | 0x372500ab | 200 |
| | ClaimRewards(address) | 0x0178be5f | 67 |
| claimReward | ClaimReward() | 0x79372f9a | 894 |
| | claimReward() | 0xb88a802f | 881 |
| | claimReward(uint256) | 0xae169a50 | 626 |
| | claimReward(uint8) | 0x689f1623 | 407 |
| | ClaimReward(address) | 0x63e32091 | 102 |
| | ClaimReward(uint256) | 0x92ceb12d | 1 |
| claimQuestRewards | claimQuestRewards(uint256[]) | 0xa7b0c81b | 528 |
| upgradeStrength | upgradeStrength(uint256) | 0xd583644b | 11 |
| getBonus | getBonus() | 0x8bdff161 | 1 |
| upgradeReward | upgradeReward(uint256) | 0x66bfdc75 | 1 |

# B DETAILS OF ALPHANUMERIC HOMOGRAPH

**Table 5: Unicode Homograph Mappings for Alphanumerics**

| Homo. | Unicode | Target | Homo. | Unicode | Target |
|-------|---------|--------|-------|---------|--------|
| Λ | U+039b | A | ∪ | U+222a | U |
| A | U+0410 | A | Χ | U+03a7 | X |
| ᗩ | U+15c5 | A | Х | U+0425 | X |
| В | U+0412 | B | у | U+0423 | Y |
| ß | U+00df | B | ¥ | U+00a5 | Y |
| Ɓ | U+0181 | B | Ζ | U+0396 | Z |
| С | U+0421 | C | α | U+03b1 | a |
| ℂ | U+2102 | C | а | U+0430 | a |
| Ð | U+0189 | D | ь | U+044c | b |
| Ε | U+0395 | E | Ƅ | U+0184 | b |
| Ɛ | U+0190 | E | ϲ | U+03f2 | c |
| ₣ | U+20a3 | F | с | U+0441 | c |
| Ƒ | U+0191 | F | δ | U+03b4 | d |
| Ғ | U+0492 | F | d | U+0501 | d |
| Ԍ | U+050c | G | ε | U+03b5 | e |
| Ǥ | U+01e4 | G | ҽ | U+04bd | e |
| Η | U+0397 | H | f | U+0192 | f |
| Н | U+041d | H | ɡ | U+0261 | g |
| ℍ | U+210d | H | ǥ | U+01e5 | g |
| Ι | U+0399 | I | η | U+03B7 | h |
| І | U+0406 | I | ι | U+03b9 | i |
| Ј | U+0408 | J | і | U+0456 | i |
| Ĵ | U+0134 | J | ¡ | U+00a1 | i |
| Κ | U+039a | K | ј | U+0458 | j |
| К | U+041a | K | κ | U+03ba | k |
| Ԟ | U+20ad | K | к | U+043a | k |
| Μ | U+039c | M | ł | U+0142 | l |
| М | U+041c | M | ɭ | U+026d | l |
| ℳ | U+2133 | M | о | U+043e | o |
| Ν | U+039d | N | τ | U+03c4 | t |
| Ŋ | U+014a | N | μ | U+03bc | u |
| О | U+041e | O | υ | U+03c5 | u |
| Ø | U+00d8 | O | ʊ | U+028a | u |
| ⵔ | U+2d54 | O | ν | U+03bd | v |
| Ο | U+039f | O | ∨ | U+2228 | v |
| Ρ | U+03a1 | P | ω | U+03c9 | w |
| Р | U+0420 | P | χ | U+03c7 | x |
| ₽ | U+20bd | P | × | U+00d7 | x |
| ℛ | U+211b | R | γ | U+03b3 | y |
| § | U+00a7 | S | ү | U+04af | y |
| Ѕ | U+0405 | S | ζ | U+03b6 | z |
| $ | U+0024 | S | ʐ | U+0290 | z |
| Τ | U+03a4 | T | ʒ | U+01b6 | z |
| Т | U+0422 | T | \| | U+007c | 1 |
| ₮ | U+20ae | T | Ƨ | U+01a7 | 2 |
| Ⴎ | U+1200 | U | ᴈ | U+0417 | 3 |
| ∪ | U+222a | U | | | |