# OpenReview forum: "Interface Illusions: Uncovering the Rise of Visual Scams in Cryptocurrency Wallets"
_ACM.org/TheWebConf/2024/Conference — TheWebConf24_

### Official Review · Reviewer_vtA4 · 2023-11-16

**Novelty:** 5
**Technical Quality:** 3

**Review:**

The paper investigates three specific types of visual deceptions in cryptocurrency wallets, providing an analytic exploration of their mechanisms and detailing detection methodologies. Running these detection strategies on Ethereum transaction data, the research delivers a measurement study of the cryptocurrency scam ecosystem, as well as conducting a revenue analysis.

The paper provides a substantial measurement study across three varied types of visual scams targeting Ethereum wallets. However, it lacks a clear definition of the threat model. For instance, in the case of the counterfeit token scam and the function name scam, the paper should provide a detailed description of the attack method: how attackers display counterfeit tokens or deceptive function names in a user's wallet. A more explanation of the threat model for each attack is needed to enhance the paper's comprehensibility.

Regarding the detection techniques, the paper lacks an analysis of the potential introduction of false positives or false negatives for the proposed methods. For the counterfeit token scam, there should be a discussion on the risk of legitimate token might exhibit characteristics that resemble counterfeit tokens when applying the forgery methods to the top-200 tokens. Similarly, in the zero-transfer scam, the paper should consider cases where legitimate addresses may coincidentally exhibit similar display patterns to deceptive ones.

While the paper presents a discussion to validate the detected results (i.e., scam cases). However, the validation process is not entirely convincing. The number of samples used for checking recall appears to be too small, and the standards employed for manually validating precision are not adequately explained. A more rigorous validation process is necessary to ensure the validity of the subsequent measurement study and its findings.

Other minor concerns include:
(1) In the function name scam detection, the paper should elaborate on the criteria used to select misleading function names during the manual inspection. Additionally, a complete list of these function names should be included to facilitate a better understanding of the analysis. There appears to be a discrepancy in Table 4 in Appendix A, where the origin of the final 17 distinct function names is noted to be from 7 instead of 8 sources.
(2) It is unclear why the paper does not provide revenue estimations for counterfeit tokens.
(3) There are some presentation issues in the paper, such as "phony effect,""elusive semantics," which should be addressed for clarity.

**Questions:**

1. Can you provide a detailed explanation of the threat models for each type of visual scam addressed in your study?

2. What was the rationale behind the sample size chosen for checking the recall of your detection methods, and could you elaborate on the standards used for manually validating precision?

**Reviewer Confidence:**

4: The reviewer is certain that the evaluation is correct and very familiar with the relevant literature

**Scope:**

4: The work is relevant to the Web and to the track, and is of broad interest to the community

---

### Official Review · Reviewer_YGcT · 2023-11-21

**Novelty:** 6
**Technical Quality:** 5

**Review:**

This study provides a comprehensive characterization and analysis of visual scams in cryptocurrency wallets through an extensive longitudinal measurement study. The authors identified approximately 24M scam incidents carried out by 6,768 attackers, resulting in a total loss of 27M USD. The research reveals the diverse aspects and dynamics of the visual scam ecosystem within cryptocurrency wallets.

+ pros
(maybe) the first attempt to characterize the visual scams in cryptocurrency wallets
timely topic
well written

- cons
lack of details in some points

Thank you for your insightful research on visual scams in cryptocurrency wallets, a topic of growing importance in the Web3 era. Your paper effectively unveils the entire ecosystem and scale of these scams, impressively structured with illustrative examples, detailed scam logic, and methodical detection approaches at the implementation level.

However, I have several suggestions that could enhance the clarity and quality of the paper:
Section 3.1 - Implementation: The method for detecting 'combo forgery' needs further details. The current description of 'testing all possible scenarios' is somewhat vague. Providing additional details, such as the set of keywords used or the strategy for adding them, would substantially clarify this section.
Section 4.1 - Timeline: For improved coherence, please reference Figure 3 directly in the main text, linking the narrative more closely with the illustrative figure.
Section 4.1 - Distribution: In Table 3, it is necessary to specify that only the top distribution, rather than the entire distribution, is presented for certain scam types. This clarification will aid in accurately interpreting the data.
Section 4.2 - Findings 3 and 4: Each scam type should be distinctly identified to ensure that the findings are not mistakenly generalized to other scam types.
Section 4.3 - Scamming Toolkits: An introduction to the toolkits employed by Zero-Transfer scammers for generating multiple Ethereum accounts resembling target addresses would be a valuable addition.
Section 4.3: There appears to be no mention of Figure 5 in the main text, nor an explanation of its contents. Integrating a reference and description of this figure would enhance understanding.
Section 4.4: The rationale for excluding the Counterfeit Token Scam from the revenue estimation is not clear. If there's a specific reason for this exclusion, please clarify it. Otherwise, for a comprehensive analysis, consider including the revenue trends of the Counterfeit Token Scam

**Questions:**

Did you try to measure/evaluate the mitigation methods proposed in the paper?

**Reviewer Confidence:**

3: The reviewer is confident but not certain that the evaluation is correct

**Scope:**

3: The work is somewhat relevant to the Web and to the track, and is of narrow interest to a sub-community

---

### Official Review · Reviewer_dG3W · 2023-11-21

**Novelty:** 4
**Technical Quality:** 4

**Review:**

This paper evaluates the feasibility and prevalence of three specific "visual scams" which deceive users of cryptocurrencies into transferring money to the wrong person. Two of them depend on a mismatch between the datum used to identify someone at the protocol level (an "address", which is the fingerprint of a public key, therefore a long string of random bits, meaningless to a human) and the datum used in the user interface (human-readable text and/or graphics, _chosen by the attacker_ -- possibly augmented with a shortened version of the address, short enough that the attacker can also select it by brute-force key generation). The third depends on the fact that there is no necessary connection between the _name_ of a software function, and what it actually does. A "smart contract" named `securityUpdate` is just as capable of stealing your money as one named `stealYourMoney`.

All of these attacks are well-known in previous literature.  The authors have done a nice job of documenting both their feasibility in this particular context, and assessing that they do indeed occur in this particular context.  If I were reviewing for a security-focused conference, or for one focused on distributed ledgers, I would not hesitate to vote for publication (with some suggestions for improvement, see "questions" section). However, I do not see _any_ relevance to the Web. It is 100% about cryptocurrency, security flaws in the user interfaces of cryptocurrency client software, and security flaws in the basic design of "smart contracts".

**Questions:**

Please explain why you submitted this paper to The Web Conference and not to Financial Crypto, Advances in Financial Technologies, IEEE S&P, USENIX Security, or any other conference that is actually about security and/or distributed ledgers.

You considered only blockchain scams and visual similarity attacks on websites in your evaluation of related work.  You therefore missed an entire subfield of previous work on visual similiarity attacks on _public key fingerprints_: for example

* PGP key presentation spoofing: "[Johnny, you are fired!](https://www.usenix.org/system/files/sec19-muller.pdf)"
* PGP short key ID spoofing: "[Evil 32](https://evil32.com/)" (short key IDs are shortened fingerprints, just like those presented by crypto wallet software)
* Visual presentation of complete fingerprints so that they aren't just "line noise": "[Can Unicorns Help Users Compare Crypto Key Fingerprints?](https://dl.acm.org/doi/abs/10.1145/3025453.3025733)"
* Petname systems and Zooko's Triangle: "[Petname Systems: Background, Theory and Applications](https://www.researchgate.net/profile/Md-Sadek-Ferdous/publication/265188437_Petname_Systems_Background_Theory_and_Applications/links/563218a908ae13bc6c371f73/Petname-Systems-Background-Theory-and-Applications.pdf)"

There has also been _some_ work on defenses against underhanded and/or mislabeled code, although [Rice's Theorem](https://en.wikipedia.org/wiki/Rice%27s_theorem) precludes any truly perfect solution. Start with "[Initial Analysis of Underhanded Source
Code](https://apps.dtic.mil/sti/pdfs/AD1122149.pdf)". Object-capability security theory might also be relevant, though I'm not aware of any work specifically on exploits that seek to deceive end users about what capabilities a blob of code possesses.

Broadening your survey to include these classes of related work would help you situate your paper and describe how wallet software might better defend against at least the first two visual scams you describe.

**Reviewer Confidence:**

3: The reviewer is confident but not certain that the evaluation is correct

**Scope:**

1: The work is irrelevant to the Web

---

### Official Review · Reviewer_b2PJ · 2023-11-22

**Novelty:** 6
**Technical Quality:** 5

**Review:**

Overall, I really enjoyed reading this paper and felt like the authors did a good job in presenting their findings in a clear and concise manner.

__Pros:__

1. They did a good job categorizing each type of attack and breaking each type of attack down into subcategories (e.g., the different types of token forgeries).

2. The dataset used in the paper is extremely large with over 24M scam incidents and 6,768 attackers.

4. The distinguish themselves from prior work by focusing largely on the visual scams associated with crypto wallets.

5. Propose mitigation approaches based on their findings.

6. At first glance, the scams demonstrated seem somewhat elementary. However, the authors were able to show the significance of these scams in their Revenue Estimation section, which showed that these attacks lead to over 27M in losses from 5,693 victims.

__Cons:__

1. It is difficult to empirically understand Figure 2, and the author's should come up with a better way to summarize this information, such as a table.

__Response Discussion:__ I appreciate the authors' comments and they provided clarification into my minor concerns. Additionally, I did update my novelty score from 5 to 6 to be better reflect my views of the new novel insights the paper provides.

**Questions:**

1. There is little justification for why the Zero-Transfer Scam went to zero starting in 23-03. It appears to be a fairly straightforward attack, so it's surprising that it stopped so suddenly. Why is this the case?

**Ethics Review Description:**

There are no ethics review needed.

**Reviewer Confidence:**

3: The reviewer is confident but not certain that the evaluation is correct

**Scope:**

4: The work is relevant to the Web and to the track, and is of broad interest to the community

---

### Official Review · Reviewer_NpVh · 2023-11-25

**Novelty:** 5
**Technical Quality:** 5

**Review:**

The paper focuses on multiple forms of cryptocurrency wallet scams in the wild by performing an analysis of the blockchain transactions. The paper identified more than 25M scam incidents, surpassing 27 million US dollar. The paper also provides an analysis of the distribution of these attacks and describes possible mitigation strategies.






**Strengths**
The paper focuses on an emerging form of scam not discussed in-depth
The dataset covers a large number of more prevalent forms of visual scams

**Weaknesses**
It would be helpful to provide more details on the threat model
The campaign analysis part needs more clarifications.


**Detailed Comments**
I would like to thank the authors for defining the project. The paper is well-written and contains several interesting findings. The idea of visual scam in crypto wallets is interesting and not very well studied in the prior work.

The paper focuses on multiple aspects of the scams in crypto wallets from the types of scams, to the monetization factor and provides interesting insights on the topics. The longitudinal analysis of the attacks were also interesting and unique.
That said, I think the paper needs to discuss a few areas in greater detail and clarify the definitions used in the paper. One example is campaign analysis. What is the definition of campaign in this context? The scams generated by the same group of adversaries?
If this is the case, how could the authors cluster different scams under one campaign? The paper might mean something else, but it is not very clear that is. It was a bit unclear to me what the authors wanted to communited without knowing the definition.

It was also not very clear to me how the paper performed an analysis of the distribution of the scam types. The number of samples acquired based on transaction history were significant. How the authors distinguished the type of scam. A more clear description on the methodology could be very helpful.

**Questions:**

Please read the reviews.

**Ethics Review Description:**

No ethical concern

**Reviewer Confidence:**

4: The reviewer is certain that the evaluation is correct and very familiar with the relevant literature

**Scope:**

4: The work is relevant to the Web and to the track, and is of broad interest to the community

---

### Decision · Program_Chairs · 2024-01-22

**Decision:**

Accept

**Comment:**

**Meta Review:**

 **Pros:**

 1. **Comprehensive Study:** The paper conducts a comprehensive analysis of visual scams in cryptocurrency wallets, providing valuable insights into the mechanisms, dynamics, and revenue aspects of these scams.

 2. **Large Dataset:** The use of a substantial dataset comprising over 24 million scam incidents and 6,768 attackers strengthens the study and contributes to the understanding of the prevalence of visual scams.

 3. **Categorization and Mitigation:** The paper categorizes different types of visual scams, including subcategories, and proposes mitigation approaches based on the findings. This contributes to practical solutions for addressing the identified issues.

 4. **Timely and Relevant:** The study addresses a timely and relevant topic in the era of Web3 and cryptocurrency, shedding light on a growing concern in the cryptocurrency wallet ecosystem.

 5. **Revenue Analysis:** The inclusion of a revenue analysis, estimating over 27 million USD in losses from 5,693 victims, provides a concrete understanding of the financial impact of visual scams.

 **Cons:**

 1. **Lack of Clear Threat Model Definition:** Reviewers express concerns about the lack of a clear definition of the threat model, particularly for specific visual scams like the counterfeit token scam and the function name scam. More detailed descriptions of attack methods are requested.

 2. **Detection Technique Analysis:** The paper is criticized for not analyzing the potential introduction of false positives or false negatives for the proposed detection methods. Specific scenarios, such as the risk of legitimate tokens resembling counterfeit tokens, should be discussed.

 3. **Validation Process Concerns:** The validation process for detected results is deemed not entirely convincing. Reviewers express reservations about the sample size used for checking recall and the lack of clear standards for manually validating precision.

 4. **Presentation Issues:** Some presentation issues, including terminology like "phony effect" and "elusive semantics," are noted, suggesting a need for improved clarity in the paper's language.

 5. **Incomplete Information and Discrepancies:** There are concerns about incomplete information, discrepancies in tables, and missing details, such as the rationale for excluding revenue estimations for counterfeit tokens.

 **Suggestions and Questions:**

 1. **Detailed Threat Model:** Clarification and detailed explanations of threat models for each type of visual scam are recommended to enhance comprehensibility.

 2. **Methodology Clarification:** A more transparent description of the methodology used for the distribution analysis and distinguishing between scam types is suggested for improved understanding.

 3. **Further Detail in Sections:** Specific sections, such as Implementation and Timeline, are suggested to include further details for clarity, coherence, and accurate interpretation of data.

 4. **Broadening Related Work:** The review recommends broadening the survey of related work to include a broader range of visual similarity attacks, such as those on public key fingerprints, to better situate the paper in the existing literature.

 5. **Explanation of Trends:** Clarification is sought for trends observed in the Zero-Transfer Scam, particularly the sudden drop in incidents in March 2023.

 **Conclusion:**

 The paper is acknowledged for its contribution to understanding visual scams in cryptocurrency wallets, but improvements in threat model definition, detection technique analysis, validation process, and presentation are recommended to enhance its overall quality and impact. Addressing these concerns could lead to a more robust and compelling contribution to the academic community.

 ---